# Nephroprotective Effect of the Herbal Composition BNO 2103 in Rats with Renal Failure

**Sergii K. Shebeko** , **Vladyslava V. Chernykh and Kateryna O. Zupanets** *

Department of Clinical Pharmacology and Clinical Pharmacy, National University of Pharmacy,
61002 Kharkiv, Ukraine; shebeko.sk@gmail.com (S.K.S.); clinpharm@nuph.edu.ua (V.V.C.)
* Correspondence: katyazupanets@gmail.com; Tel.: +38-067-917-4273

**Abstract:** (1) Background: this research aims at studying the nephroprotective properties of BNO 2103 in a model of chromate-induced renal failure in rats and at proving the possibility of using BNO 2103 in clinical practice for the complex treatment of chronic kidney disease (CKD). (2) Methods: fifty rats divided into five groups were studied. The drugs BNO 2103, Prednisolone and Lespephril were administered within 20 days. The excretory function and the functional state of kidneys, blood biochemical parameters and indicators of nitrogen metabolism were determined. (3) Results: under the influence of BNO 2103, there was a significant improvement in renal excretory function, in nitrogen metabolism and blood biochemical parameters compared with the control pathology group. BNO 2103 also outperformed the comparators in most indicators. (4) Conclusions: BNO 2103 has demonstrated nephroprotective, hypoazotemic and diuretic effects; and can be used to implement to the combined therapy of CKD.

**Keywords:** chronic kidney disease; BNO 2103; nephroprotective effect; phytotherapy; experimental treatment

## 1. Introduction

Chronic kidney disease (CKD) is one of the global problems of today's world medicine [1]. Actualizing the issue of the CKD incidence is currently vital, as this problem results in both socio-economic and general medical aspects [2].

According to the study conducted in twelve countries with a total number of 75,058 participants, the prevalence of CKD among adults is quite high: 14.3% in the general population, 36.1% in high-risk groups (hypertension, diabetes, cardiovascular disease) [3]. About two million people suffer from end-stage renal disease and the percentage of such diseases increases by 5–7% annually [4]. According to the World Health Organization, each year, from five to ten million people die from kidney disease [5].

Although only about 1% of patients with CKD require dialysis or kidney transplantation, the economic costs of dialysis and transplantation are very high in all countries—on average, 5% of the annual health budget is spent on 1% of the population [6]. In this regard, it is extremely important to identify patients with CKD timely, in particular in the first stages of medical care, as well as to provide prevention and treatment as early as possible.

There is an extensive empirical and experimental experience in the use of phytoneering drugs in the treatment of kidney and urinary system diseases. However, the evidence base is still insufficient. Phytopharmaceuticals are used successfully for the symptomatic treatment of CKD but they are presented in a rather narrow range despite a number of advantages, including mild therapeutic effect and a much smaller list of side effects. Thus, the current practical task today is to find ways to expand the indications for already known phytotherapeutic agents and study the possibility of their use for the treatment of CKD.

Known in clinical practice, the combined herbal remedy BNO 2103 (Bionorica SE, Germany) is used in the complex treatment of cystitis and to prevent urolithiasis [7,8]. Such indications for use are due to the composition of the cure—an equal combination of centaury grass, rosemary leaves and lovage root. This remedy does not only have a diuretic effect but also antibacterial, antioxidant, antispasmodic and anti-inflammatory activities. The nephroprotective activity, anti-inflammatory and analgesic effect of the drug was proven in experimental Heymann nephritis [9], cystitis [10] and prostatitis models [11]. The antioxidant activity of BNO 2103 in diabetic nephropathy has also been proven in a clinical study [12]. In a study involving patients with steatohepatitis and stage I-II CKD, there is evidence of a significant improvement in renal excretory function on the background of the combined use of BNO 2103 and S-adenosylmethionine [13]. The possibility of the safe use of the drug in pregnant women [14–16] and in the case of allergy to antibacterial drugs has also been proven [17]. At the same time, to date, this number of modern studies is not enough to form a sufficient evidence base that would justify the use of BNO 2103 in CKD.

The aim of this research is to study the nephroprotective properties of BNO 2103 using a model of renal failure (RF) in rats and to prove the possibility of its use in clinical practice for the complex treatment of CKD.

## 2. Materials and Methods

### 2.1. Animals

The experimental study was performed using 52 random-bred male albino rats, weighing 180–200 g, which were obtained from the vivarium of the Central Research Laboratory of the Educational and Scientific Institute of Applied Pharmacy (CRL ESIAPh) of National University of Pharmacy (NUPh) (Kharkiv, Ukraine). The animals received a standard rat diet and water ad libitum. The rats were housed under standard laboratory conditions in a well-ventilated room at 25 ± 1 °C and a relative humidity 55 ± 5% with a regular 12 h light/12 h dark cycle [18,19]. All the studies were conducted in accordance with EU Council Directive 2010/63/EU on compliance with the laws, regulations and administrative provisions of the EU Member States on the protection of animals used for scientific purposes [20]. The experimental protocols were approved by the Bioethics Commission of the NUPh (Approval No. 2 dated 4 November 2019).

### 2.2. Study Object and Its Preparation

The object of the study was the combined phytocomposition BNO 2103 (study code PM 19-016), which was provided by "Bionorica SE" (Neumarkt, Germany). The studied phytocomposition comprises the mixture of such extracts as centaury grass, rosemary leaves and lovage root in the form of finely dispersed hygroscopic brown powder with a specific odor of vegetable raw materials. Immediately before administration, the corresponding suspension was made from the powder using a vehicle containing 0.1% Tween and 0.5% Cekol mixed by vortex V-1 plus (Biosan, Riga, Latvia).

### 2.3. Comparators and Their Preparation

Prednisolone 5 mg in tablets (Darnitsa, Kyiv, Ukraine) was chosen as the first comparator as an anti-inflammatory, immunosuppressive, antiproliferative agent widely used in CKD associated with glomerulopathies and glomerulosclerosis [21]. This object was introduced in the form of a suspension prepared similarly to BNO 2103.

An oral solution Lespephril (Lubnypharm, Lubny, Ukraine), a herbal drug of diuretic and hypoazotemic action, containing Lespedeza extract was used as the second comparator. Lespedeza herb tinctures are widely used in RF [22]. This object was introduced in its original form without modification.

### 2.4. Research Design

All animals were randomly divided into 5 experimental groups as follows.

Group 1—intact control (IC) (healthy animals receiving the vehicle, *n* = 10).
Group 2—control pathology (CP) (untreated animals receiving the vehicle, *n* = 12).
Group 3—animals with RF treated with BNO 2103 at 33.0 mg/kg (*n* = 10).
Group 4—animals with RF treated with Prednisolone at 1.9 mg/kg (*n* = 10).
Group 5—animals with RF treated with Lespephril at 3.0 mL/kg (*n* = 10).

Chromium-induced nephropathy was used as an RF model [23]. It was induced by a single subcutaneous injection of 2.5% potassium chromate solution (Sigma-Aldrich, St. Louis, MO, USA) at a dose of 0.7 mL/kg in the original modification as described earlier [24] on the first day of the experiment. Starting from the second day of the experiment, the animals received BNO 2103 at a dose of 33.0 mg/kg (the average therapeutic dose for humans extrapolated according to FDA recommendations [25]), Prednisolone at a dose of 1.9 mg/kg (ED50 for anti-inflammatory activity) and Lespephril at a dose of 3.0 mL/kg (*n* = 10) (the average therapeutic dose for humans extrapolated according to FDA recommendations). All the test samples were administered intragastrically daily for 20 days as suspensions prepared in the vehicle. The animals of the control groups received an equivalent amount of the vehicle. On completion of the study, the animals were sacrificed under general anesthesia with ketamine/xylazine (75/10 mg/kg, intraperitoneally) [26] to obtain blood for biochemical assays and renal tissue for histological evaluation.

## 2.5. Biological Samples Preparation and Storage

The blood samples were collected from the inferior vena cava and were centrifuged at 1500× *g* at 4 °C for 10 min using refrigerated centrifuge "Eppendorf 5702R" ("Eppendorf AG", Hamburg, Germany). The urine samples were collected using individual metabolic cages and were centrifuged at 500× *g* for 10 min. The supernatants were separated and used for the biochemical assays. All the blood and urine samples were frozen and stored at −80 °C until use. The kidneys were quickly removed, subjected to macroscopic analysis and weighed. Their weight was used for kidney weight coefficient (KWC) calculation relative to body weight. Then kidneys were fixed in 10% neutral-buffered formalin and embedded in paraffin blocks for histology test.

## 2.6. Evaluation of the Functional State of Kidneys

Spontaneous daily diuresis was determined with individual metabolic cages at the end of the experiment in all the animals. The protein content and its daily excretion were determined in the collected urine. The glomerular filtration rate was evaluated as endogenous creatinine clearance (CC), tubular reabsorption (TR) and urea clearance (UC) were also calculated, using the standard equations [27]:

$$CC = Ucr \times V/Pcr \tag{1}$$

$$TR = (1 - Pcr/Ucr) \times 100\% \tag{2}$$

$$UC = Uur \times V/Pur \tag{3}$$

where Ucr is the urine creatinine concentration, V is the daily diuresis, Pcr is the plasma creatinine concentration, Uur is the urine urea concentration and Pur is the plasma urea concentration.

## 2.7. Biochemical Assays

The biochemical studies to evaluate the parameters of the renal excretory function, nitrogen and protein metabolism were performed using the appropriate biochemical commercial kits (High Technology, Inc., North Attleboro, MA, USA) and the "Express Plus" automatic biochemical analyzer (Bayer Diagnostics, Leverkusen, Germany). Creatinine and urea levels in blood and urine were determined using a kinetic test without deproteinization according to the Jaffe method and urease glutamate dehydrogenase enzymatic UV test, respectively [28]. The urinary excretion of creatinine and urea were also calculated. The total blood protein was determined photometrically by biuret method, the albumin content—by reaction with

bromocresol green. The protein urine concentration was determined by the pyrogallol red-molybdate method [29].

## 2.8. Histological Examination

Paraffin sections (5 μm thick) were mounted on slides and stained with periodic acid-Schiff (PAS). The sections were examined under the light microscope "Optika B-1000BF" and photographed with the digital camera "Optikam HDMI Pro" (Optika, Bergamo, Italy) [30]. To determine the degree of glomerulosclerosis, a semi-quantitative score was obtained from 10 glomeruli on PAS-stained sections by multiplying the grades for mesangial expansion, dilation of Bowman's capsule and segmental and global sclerosis (grade 0—no damage; 1—damage in 25%; 2—damage in 50%; 3—damage in 75%; 4—damage in the whole area of the glomeruli in the visual field). Tubular damages were assessed on PAS-stained sections by scoring tubular cell necrosis, tubular dilatation, hyaline cast deposition and brush border loss in 10 non-overlapping fields in the cortex and corticomedullary junction. Injury was scored by on a 5-point scale (grade 0—no damage; 1—damage in 1–10%; 2—damage in 10–25%; 3—damage in 25–50%; 4—damage in 50–75%, 5—more than 75%). All the results of semi-quantitative scores were averaged [31].

## 2.9. Statistical Analysis

All the results were processed by descriptive statistics and presented as the mean ± standard error of the mean (SEM). The statistical differences between groups were analyzed using one-way ANOVA followed by Tukey post hoc test [32]. The computer software used included IBM SPSS Statistics v. 22 (IBM Corp., Armonk, NY, USA) and MS Excel 2016 (Microsoft Corp., Redmond, WA, USA). The level of statistical significance was considered as $p < 0.05$.

## 3. Results

### 3.1. Effect of BNO 2103 on Kidney Function in Rats with Renal Failure

The results of the study demonstrated that in the CP group under the influence of potassium chromate, there was a significant RF, which manifested itself in 25% of animals' mortality, the reduction in body weight ($p < 0.05$ relative to IC), diuresis, TR and a 2.8-fold decrease in CC. At the same time, there has been the expressed proteinuria which reached 23.0 mg/day observed (Table 1).

**Table 1.** Effect of BNO 2103 on the course of renal failure in rats.

| Groups of Animals | Body Weight (g) | Diuresis (mL/Day) | Proteinuria (mg/Day) | Creatinine Clearance (mL/Day) | Tubular Reabsorption (%) |
|---|---|---|---|---|---|
| Intact control ($n = 10$) | 196.0 ± 3.8 | 5.3 ± 0.4 | 1.5 ± 0.1 | 553.9 ± 9.4 | 99.04 ± 0.08 |
| Control pathology ($n = 9$) | 168.2 ± 3.3 [a] | 4.1 ± 0.2 [a] | 23.0 ± 0.8 [a] | 197.0 ± 4.8 [a] | 97.92 ± 0.12 [a] |
| BNO 2103 33.0 mg/kg ($n = 10$) | 201.9 ± 2.5 [b] | 5.7 ± 0.3 [b] | 14.9 ± 0.3 [a,b] | 332.0 ± 10.7 [a,b] | 98.28 ± 0.06 [a,b] |
| Prednisolone 1.9 mg/kg ($n = 9$) | 211.4 ± 5.3 [a,b] | 4.2 ± 0.3 [a,c] | 13.6 ± 0.7 [a,b] | 229.2 ± 9.7 [a,b,c] | 98.17 ± 0.09 [a] |
| Lespephril 3.0 mL/kg ($n = 9$) | 208.0 ± 3.6 [a,b] | 6.5 ± 0.4 [a,b] | 17.1 ± 0.8 [a,b,c] | 310.0 ± 13.6 [a,b] | 97.88 ± 0.14 [a,c] |

Data are expressed as mean ± SEM ($n$ is the number of animals at the end of experiment); [a] $p < 0.05$ vs. intact control group; [b] $p < 0.05$ vs. control pathology group; [c] $p < 0.05$ vs. group treated with BNO 2103.

While using the study object BNO 2103 in the animals with RF the mortality disappeared, and there was a significant increase in the body weight and excretory function of the kidneys compared to the untreated animals. This was manifested in the diuresis increase ($p < 0.05$), the TR, a 1.5-fold

decrease in the proteinuria and 1.7-fold increase in the CC (Table 1). Under the influence of the reference drug Prednisolone, there a lower level of overall efficacy than with BNO 2103 was observed. Prednisolone was expected to show a pronounced antiproteinuric effect, with a 1.7-fold proteinuria decrease, but it increased the CC by only 16.3% ($p < 0.05$) (Table 1). The second comparator Lespephril showed a pronounced diuretic effect, increasing the diuresis and CC 1.6 times ($p < 0.05$). Additionally, under its influence, there was a 1.3-fold decrease ($p < 0.05$) in the proteinuria (Table 1). Compared to Prednisolone, BNO 2103 increased the diuresis by 35.7% and the CC by 44.9% ($p < 0.05$). BNO 2103 has been expected to be inferior to Lespephril in its diuretic activity, but it was 86.9% more effective in reducing the level of proteinuria ($p < 0.05$).

### 3.2. Effect of BNO 2103 on Residual Nitrogen Excretion in Rats with Renal Failure

The deterioration of the functional state of the kidneys in the CP group led to a decrease in the excretion of nitrogenous compounds and an increase in the content of residual nitrogen in the blood of the animals. Thus, the creatinine and urea excretion decreased ($p < 0.05$) by 16.6% and 21.9%, respectively, and the UC decreased threefold (Figures 1 and 2). The content of creatinine and urea in the blood increased ($p < 0.05$) 2.3–2.4 times compared with the IC group (Table 2).

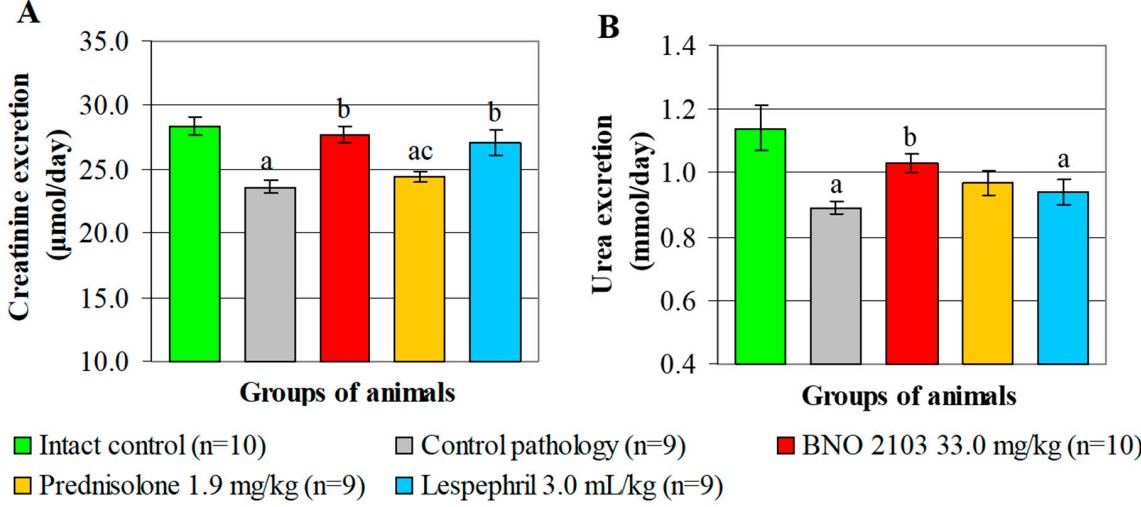

**Figure 1.** Urinary excretion of creatinine (**A**) and urea (**B**) under the influence of BNO 2103 in rats with renal failure. Data are expressed as mean ± SEM (*n* is the number of animals at the end of experiment); [a] $p < 0.05$ vs. intact control group; [b] $p < 0.05$ vs. control pathology group; [c] $p < 0.05$ vs. group treated with BNO 2103.

Under the influence of BNO 2103, positive changes occurred in the indicators of nitrogen metabolism. The study drug increased ($p < 0.05$) the urinary excretion of creatinine by 17.4%, the urea by 15.7% and the UC by 1.7-fold, correspondingly (Figures 1 and 2). As a result, the creatinine and blood urea decreased ($p < 0.05$) by 1.4–1.5 times, which indicates a decrease in the manifestations of azotemia and normalization of nitrogen metabolism. Accordingly, Prednisolone increased the UC by 19.0% and had no probable effect on the excretion of nitrogen compounds (Figures 1 and 2). As a result, the blood creatinine increased by only 10.5% ($p < 0.05$), and the urea changed insignificantly. Lespephril had a moderate hypoazotemic effect, increasing ($p < 0.05$) the creatinine excretion by 14.9% and the UC by 43.0% (Figures 1 and 2), while the creatinine and blood urea decreased 1.4-fold. Compared to Prednisolone, BNO 2103 increased the UC by 45.3% ($p < 0.05$). BNO 2103 also exceeded prednisolone ($p < 0.05$) by 21.9% and 26.0% in terms of the effect on the creatinine and blood urea, respectively. Additionally, BNO 2103 increased the UC level 22.0% more significantly than Lespephril ($p < 0.05$).

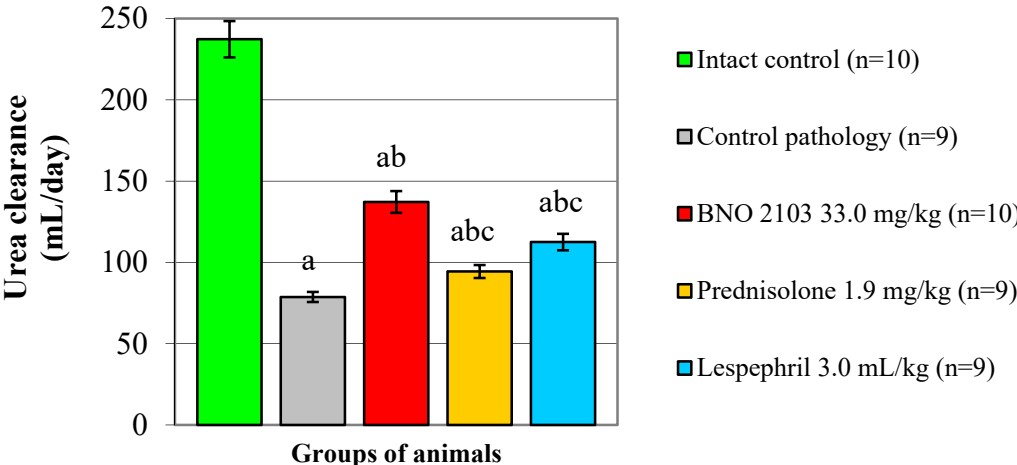

**Figure 2.** Influence of BNO 2103 on the urea clearance in rats with renal failure. Data are expressed as mean ± SEM (*n* is the number of animals at the end of experiment); [a] $p < 0.05$ vs. intact control group; [b] $p < 0.05$ vs. control pathology group; [c] $p < 0.05$ vs. group treated with BNO 2103.

**Table 2.** Biochemical blood parameters of rats with renal failure under the influence of BNO 2103.

| Groups of Animals | Creatinine (μmol/L) | Urea (mmol/L) | Total Protein (g/L) | Albumin (g/L) |
|---|---|---|---|---|
| Intact control (*n* = 10) | 51.2 ± 1.2 | 4.8 ± 0.2 | 70.5 ± 1.0 | 28.3 ± 0.4 |
| Control pathology (*n* = 9) | 120.2 ± 3.7 [a] | 11.5 ± 0.5 [a] | 61.7 ± 0.8 [a] | 20.3 ± 0.6 [a] |
| BNO 2103 33.0 mg/kg (*n* = 10) | 84.0 ± 2.2 [a,b] | 7.6 ± 0.2 [a,b] | 67.7 ± 0.8 [a,b] | 26.0 ± 0.5 [a,b] |
| Prednisolone 1.9 mg/kg (*n* = 9) | 107.6 ± 3.5 [a,c] | 10.3 ± 0.3 [a,c] | 65.9 ± 0.5 [a,b] | 27.4 ± 0.5 [b] |
| Lespephril 3.0 mL/kg (*n* = 9) | 87.9 ± 2.8 [a,b] | 8.4 ± 0.4 [a,b] | 63.2 ± 0.6 [a,c] | 22.5 ± 0.8 [a,c] |

Data are expressed as mean ± SEM (*n* is the number of animals at the end of experiment); [a] $p < 0.05$ vs. intact control group; [b] $p < 0.05$ vs. control pathology group; [c] $p < 0.05$ vs. group treated with BNO 2103.

### 3.3. Effect of BNO 2103 on Blood Protein Content in Rats with Renal Failure

As a result of the persistent proteinuria in the CP group, there was a decrease ($p < 0.05$) in the total protein and blood albumin by 12.5% and 28.3%, respectively, which indicates the selective nature of proteinuria (Table 2). Under the influence of BNO 2103, levels of total protein and blood albumin were increased by 9.7% and 28.0%, respectively ($p < 0.05$), which is due to the antiproteinuric action of BNO 2103 (Table 2). Under the antiproteinuric influence of Prednisolone, there was an increase ($p < 0.05$) in the total protein and blood albumin by 6.8% and 35.0%, respectively (Table 2). However, under the influence of Lespephril, there was no significant effect on the protein excretion and metabolism (Table 2). BNO 2103 was 86.9% more effective in reducing the level of proteinuria and increased the blood albumin 15.6% more significantly than Lespephril ($p < 0.05$).

### 3.4. Effect of BNO 2103 on Kidney Weight and Structure

As a result of active inflammatory and destructive processes in the CP group, there was a significant increase in KWC by 42.0% ($p < 0.05$). Under the influence of BNO 2103, this indicator decreased to the IC level ($p < 0.05$), while the influence of the test object was approximately the same as that of the comparators without a significant difference (Figure 3).

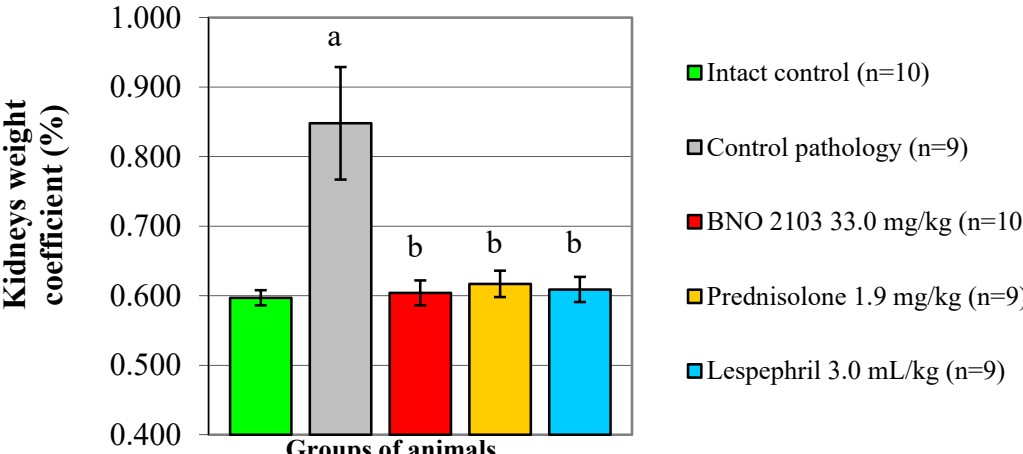

**Figure 3.** Influence of BNO 2103 on the kidneys' weight coefficient in rats with renal failure. Data are expressed as mean ± SEM (*n* is the number of animals at the end of experiment); [a] $p < 0.05$ vs. intact control group; [b] $p < 0.05$ vs. control pathology group.

Histological examination of renal tissue in the CP group (Figure 4B) revealed significant pathological changes in comparison with IC (Figure 4A). In this case, both glomeruli (thickening of the parietal lamina and expansion of the urinary space of Bowman's capsule, proliferation of the mesangium, development of glomerular sclerosis) and the tubular zone (expansion of the lumen of the tubules, accumulation of hyaline cylinders in them, loss of the brush border of tubular epithelium and its desquamation) were affected. The score for glomerusclerosis was 3.3 (Figure 4F), and for tubular lesions—4.3 (Figure 4G). BNO 2103 conduced to the preservation of cytoarchitectonics of the renal tissue of both glomerular and tubular nephron components. Under its influence, the thickness of the Bowman's capsule wall and the number of mesangial cells decreased, the brush border of the epithelial cells of the tubules was stained well, and hyaline deposits were practically not found (Figure 4C). The rate of glomerular sclerosis decreased by 39.4%, and that of tubular lesions by 48.8% ($p < 0.05$). The comparator prednisolone had the same level of effect on glomeruli as BNO 2103, but was inferior in its effect on the tubular zone of the renal tissue (Figure 4D), while under its influence, the rate of tubular lesion was 36.4% higher ($p < 0.05$). Lespephril also had a positive effect on kidney tissue, especially on the tubules, but it had a poorer effect on nephron glomeruli compared to BNO 2103. At the same time, there was a thickening of the outer layer of Bowman's capsule and active proliferation of mesangial cells (Figure 4E); the glomerulosclerosis index was 2.9, which was 45% higher than with BNO 2103 ($p < 0.05$) and did not differ from the CP group (Figure 4F).

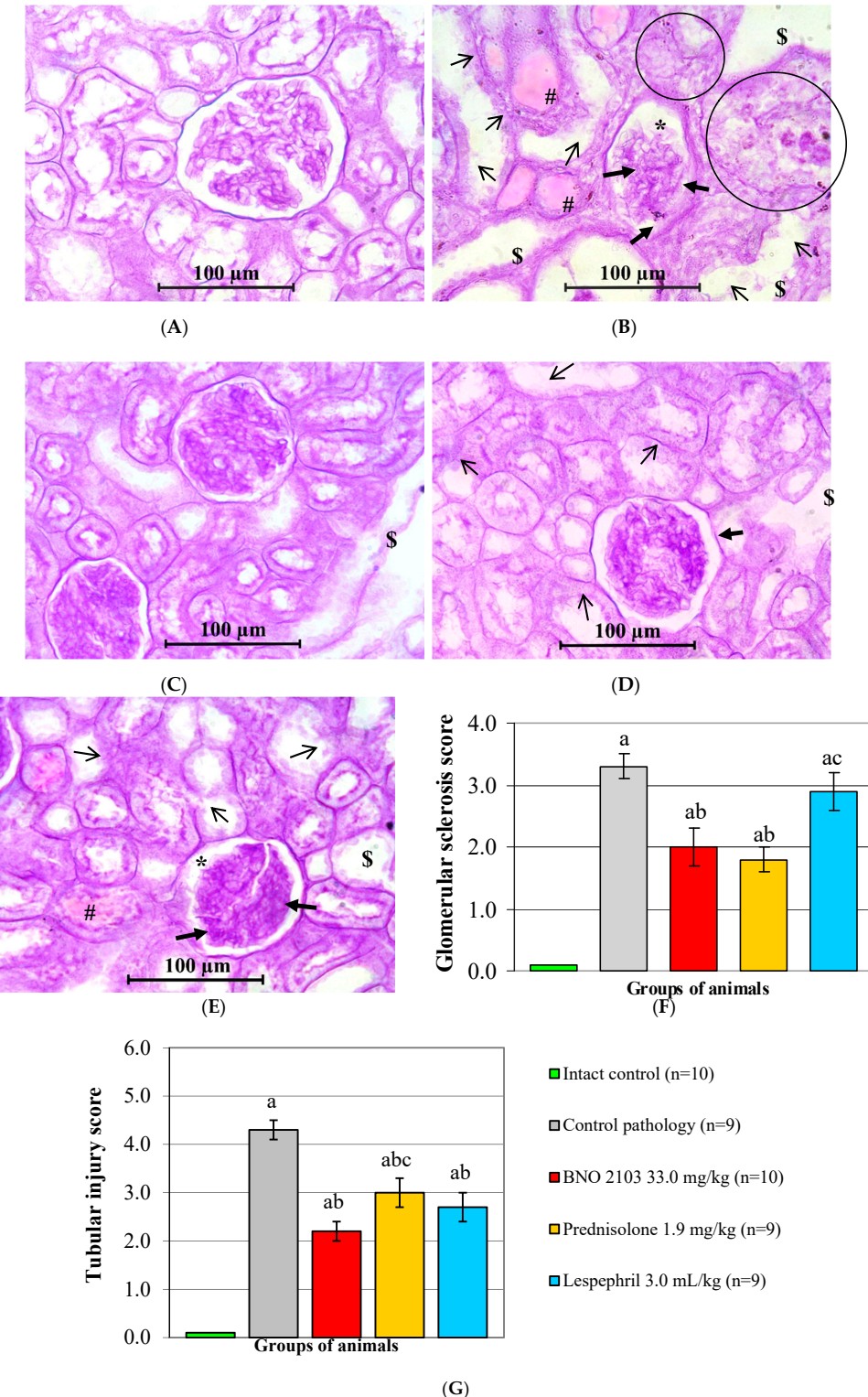

**Figure 4.** Histological examination of kidneys under the influence of BNO 2103. Representative light micrographs of renal tissues obtained from intact control group (**A**), control pathology group (**B**), animals receiving BNO 2103 (**C**), animals receiving Prednisolone (**D**), animals receiving Lespephril (**E**); mesangial expansion (thick arrows), dilation of Bowman's capsule (*), tubular cell necrosis (circles), tubular dilatation ($), hyaline casts (#), brush border loss (thin arrows); PAS staining (x250). The graph showed the semi-quantitative measurement of glomerular sclerosis scores (**F**) and tubular injury scores (**G**). Data are expressed as mean ± SEM (*n* is the number of animals at the end of experiment); [a] $p < 0.05$ vs. intact control group; [b] $p < 0.05$ vs. control pathology group; [c] $p < 0.05$ vs. group treated with BNO 2103.

## 4. Discussion

The results of the study have proved that the complex phytoneering drug BNO 2103 has a pronounced positive effect on the course of RF. When used, there is an improvement in the physical condition of rats and an increase in their survival to 100% (against 75% in the group of CP). At the same time, in rats there was an increase in the excretory kidney function; a probable increase in diuresis, in the intensity of glomerular filtration and tubular reabsorption; and a proteinuria decrease. This led to an increase in the excretion of residual nitrogen in the urine and a decrease in its content in the blood. Moreover, due to the antiproteinuric action of BNO 2103, there was an increase in the content of the total protein in the blood mainly due to albumin, which is explicated not only by the decrease in urinary protein excretion, but also by an increase in the selectivity of proteinuria. In general, all the above indicates an increase in the resistance of nephrons to adverse conditions of pathologically altered kidneys, the restoration of their filtration and reabsorption function, which, in general, reveals a nephroprotective effect. The high objectivity of the obtained results is evidenced by the data of histological assessment, which confirm the nephroprotective properties of BNO 2103, and indicate the good preservation of the cytoarchitectonics of the renal tissue under its influence, a decrease in the score of glomerulosclerosis and tubular damage. The hypoazotemic effect of BNO 2103 is explained by the improvement of intraglomerular hemodynamics, which leads to an increased excretion of nitrogen metabolism products in the urine under the influence of filtration processes.

The above-described pharmacological properties of BNO 2103 are determined by a complex of biologically active substances (flavonoids, phenolic acids, essential oils) contained in the plant raw material of the study object—centaury grass, rosemary leaves and lovage roots [33,34]. The key pharmacological effect underlying the nephroprotective properties of BNO 2103 is probably anti-inflammatory. It is known that rosemary acid, which is the main active ingredient of BNO 2103, inhibits the non-specific activation of both complement and lipoxygenase, as a result inhibiting the synthesis of leukotrienes, and breaking the chain of radical reactions [35]. The anti-inflammatory properties of BNO 2103 have also been observed in a number of the previously conducted experimental and clinical studies, where they led to an overall positive effect on the course of cystitis [10,11,17].

The obtained results correlate with the results of the study [12], which manifested the nephroprotective effect of BNO 2103, which expressed itself by a decrease in the permeability of glomerular capillaries and antiproteinuric effect in patients with diabetic nephropathy. Similar conclusions about the nephroprotective and anti-inflammatory properties of BNO 2103 were obtained in an experimental study on a model of Heymann nephritis [9]. In our experiment, the phytoneering composition BNO 2103 also showed a significant effect on the improvement of renal filtration processes, which corresponds to the results of a clinical study in patients with a combination of CKD stage I-II and steatohepatitis [13].

When comparing the efficacy of the study object with the comparators, it was found that BNO 2103 probably exceeded the efficacy of Prednisolone in terms of the effects on renal excretory function, nitrogen elimination and blood levels. It also significantly exceeded the efficacy of Lespephril in terms of its effect on proteinuria, UC and protein metabolism. In this connection, the pharmacodynamic spectrum of BNO 2103 combines the most useful effects of the comparators—nephroprotective, hypoazotemic, anti-inflammatory, antiproteinuric, diuretic ones. Thus, BNO 2103 has more balanced pharmacological properties for the treatment of renal pathology than the comparison drugs.

The importance and relevance of this study is to expand the indications of the already known phytomedicine BNO 2103 in order to increase the treatment of patients with CKD who need an effective, timely and safe therapy. CKD is a multifaceted disease, whose pathogenesis varies, depending on the primary pathology, but in any case, the most frequent result is RF. For that matter, in the present study, the appropriate experimental model was chosen. In this work, the issues of combined therapy of BNO 2103 with other drugs for the treatment of CKD and the use of phytoneering drug for the treatment of CKD in combination with other diseases were not considered. Additionally, the results of this research are based on biochemical analyses and functional indicators of kidneys and laboratory

observation. Due to this, eventually, we will perform a histomorphological and immunohistochemical evaluation of the effectiveness of BNO 2103 in order to obtain more in-depth evidence data.

## 5. Conclusions

Under conditions of the RF development in rats, the studied phytoneering composition BNO 2103 improves the physical condition of animals, reduces their mortality, enhances the renal excretory function, normalizes nitrogen and protein metabolism, contributes to protection of the kidney tissue structure and, therefore, has a positive effect on the course of nephropathy. At the same time, BNO 2103 is not only equal but, according to some indicators, superior as for the effectiveness of comparator drugs such as Prednisolone and Lespephril. Thus, in experimental studies, BNO 2103 has a pronounced nephroprotective and hypoazotemic effect and is a promising tool for the treatment of CKD. Furthermore, much remains to be learned about the exact mechanism of action of BNO 2103 as well as immunostaining for kidney tissue apoptosis and markers of podocyte inflammation in case of RF.

It is advisable that BNO 2103 should be further studied in preclinical and clinical trials in order to expand the evidence base, as well as to search for possible safe and effective combinations with drugs that potentiate the effect of BNO 2103, which can be considered as a core combined therapy for renal pathology.

**Author Contributions:** Conceptualization, S.K.S. and K.O.Z.; methodology, S.K.S.; software, S.K.S.; validation, S.K.S. and K.O.Z.; formal analysis, S.K.S. and V.V.C.; investigation, S.K.S. and V.V.C.; resources, S.K.S.; data curation, S.K.S.; writing—original draft preparation, V.V.C.; writing—review and editing, S.K.S. and K.O.Z.; visualization, S.K.S.; supervision, K.O.Z.; project administration, K.O.Z.; funding acquisition, K.O.Z. All authors have read and agreed to the published version of the manuscript.

**Funding:** This research was funded by Bionorica SE.

**Acknowledgments:** The authors are grateful to the staff and the heads of the CRL ESIAPh (Kharkov, Ukraine), Department of Clinical Pharmacology and Clinical Pharmacy of NPhU (Kharkov, Ukraine), Clinical Diagnostics Laboratory (Kharkiv, Ukraine) and Head of the Department of Clinical Pharmacology and Clinical Pharmacy, I.A. Zupanets for providing the material and technical base for the study.

**Conflicts of Interest:** The authors declare that there is no conflict of interest. The funders had no role in the design of the study; in the collection, analyses, or interpretation of data; in the writing of the manuscript, or in the decision to publish the results.

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
