# Peer review of "Nephroprotective Effect of the Herbal Composition BNO 2103 in Rats with Renal Failure"

_scipharm, doi:10.3390/scipharm88040047_

Round 1
Reviewer 1 Report
The authors demonstrate clearly that BNO 2103 probably exceeded the efficacy of Prednisolone in terms of the effects on renal excretory function, nitrogen elimination and blood levels. I thought it is very interest and valuable.
Minor
- The authors should describe clearly how did they calculate the sample size.
- Please explain why the sample size described in the method part (line 90~94) and the number the authors obtained as the result were different. Was there a bias in deleting data?
Author Response
The authors want to convey appreciation for taking time and efforts to the respective reviewer for providing such an insigtful guidance. Please see the attachment.

Reviewer 2 Report
In this manuscript, authors demonstrated that BNO 2103, a combined herbal remedy which has been used for cystitis treatment and urolithiasis prevention, improved renal function and renal histology in a rat model of chromate-induced renal failure. The subject of study seems to be interesting. However, there are some concerns in this study. The reviewer’s comments are described as follows.
1. The most serious issue in this study is that mechanisms by which BNO 2103 ameliorated renal injury in rat model of renal failure remained unclear. As authors described in In Introduction and Discussion, BNO 2103 possesses anti-inflammatory and anti-oxidative effects according to some previous studies. In addition, authors used prednisolone as positive control of anti-inflammatory agent. Therefore, authors should clearly demonstrate anti-inflammatory and anti-oxidative effects of BNO 2103 in this animal model by measurement of oxidative stress markers such as 8-OHdG in urine and inflammatory infiltration into glomeruli and tubulointerstitial area by immunostaining.
2. Since BNO 2103 reduced proteinuria, authors should evaluate not only glomerulosclerosis but also podocyte injury based on immunostaining for podocyte proteins.
3. Renal tubular injury was insufficiently evaluated. Tubular injury including necrosis, dilation, hyaline casts, and brush border loss should be indicated by arrows in figures. Furthermore, since tubular injury score generally has poor quantitative property, renal tubular injury should be examined more specifically, for example by immunostaining for kidney injury molecule (KIM)-1 or TUNEL staining for apoptosis.
4. Since creatinine clearance cannot be a surrogate marker for glomerular filtration rate (GFR), GFR in text and tables should be replaced with creatinine clearance.
5. In Table 1, blood pressure and body weight are essential parameters for animal models of chronic kidney disease and renal failure. Authors should add these parameters to this table.
Author Response
The authors want to convey appreciation for taking time and efforts to the respective reviewer for providing such an insigtful guidance. Please see the attachment

Round 2
Reviewer 2 Report
Authors have successfully responded the reviewer's comments. However, regarding the comment #1, #2, and #3, authors' responses and explanations should be reflected in the revised manuscript.
Author Response
Dear reviewer!
I would like to kindly inform you that we have made the corresponding corrections in the article attached:
1) comment #1 : added to line #49, lines #331-333
2) comment #2 : added to lines #331-333 that reflect our interests in future in-depth studies
3) comment #3: lines #231-244 contain the arrows which are presented in figures along with explanation in the legend.
Thank you so much for your suggestions and comments.
Kindest regards,
Dr. Kate Zupanets
National university of Pharmacy